# Coordination Chemistry of Phosphate Groups in Systems Including Copper(II) Ions, Phosphoethanolamine and Pyrimidine Nucleotides

**DOI:** 10.3390/ijms232213718

**Published:** 2022-11-08

**Authors:** Malwina Gabryel-Skrodzka, Martyna Nowak, Anna Teubert, Renata Jastrzab

**Affiliations:** 1Faculty of Chemistry, Adam Mickiwicz University in Poznan, Uniwersytetu Poznanskiego 8, 61-614 Poznań, Poland; 2Institute of Bioorganic Chemistry, Polish Academy of Sciences, Noskowskiego 12/14, 61-704 Poznań, Poland

**Keywords:** coordination chemistry, copper(II) ion complexes, pyrimidine nucleotides, phosphoethanolamine, complexes

## Abstract

The activity of phosphate groups of phosphoethanolamine and pyrimidine nucleotides (thymidine 5-monophosphate, cytidine 5-monophosphate and uridine 5’monophosphate) in the process of complexation metal ions in aqueous solution was studied. Using the potentiometric method with computer calculation of the data and spectroscopic methods such as UV-Vis, EPR, ^13^C and ^31^P NMR as well as FT-IR, the overall stability constants of the complexes as well as coordination modes were obtained. At lower pH, copper(II) ions are complexed only by phosphate groups, whereas the endocyclic nitrogen atom of nucleotides has been identified as a negative center interacting with the -NH_3_^+^ groups of phosphoethanolamine.

## 1. Introduction

The binding of metal ions by bioligands and the interactions between them have been the subject of study for years. Nucleic acids are especially important here, and their behavior has been studied most often. Metal ions occurring in living systems form coordination bonds with, e.g., DNA or RNA bases and phosphate groups. Depending on the type of metal and the sequence of nucleotides, the binding of the metal to the bases could destabilize the double helix structure [1,2,3,4]. On the other hand, metal ions are also involved in stabilizing the DNA structure, such as in coordination with the negatively charged phosphate backbone [5,6,7]. Research on the coordination chemistry of nucleic acids and d-block metal ions, such as copper, is extremely interesting—they could lose their water molecules very easily and give inner sphere coordinated complexes [8].

Copper ions are an important microelement in the human body and play a role as a co-factor for enzymes such ascytochrome oxidase, ferrooxidases, superoxide dismutase, and amine oxidases. The total amount of copper ions in the human organism, not only in blood but also in individual organs, is in homeostatic control. Changes in these amounts may indicate disorders and diseases. Excessive amounts of copper have the potential to damage cells and their constituents, especially due to the production of reactive oxygen species and DNA and chromatin damage, and can be the basis for the effects of copper related to cancer and other pathologies [9,10,11,12,13]. Copper complexes are also promising for anticancer treatment—they can interact with DNA, and the mechanism of action is still the subject of research by scientists [14,15,16]. They predominantly form non-covalent interactions with DNA via electrostatic forces of attraction, intercalation or minor groove binding [17,18,19]. For this reason, it is extremely important to study the interactions of metal ions, such as copper(II), especially with phosphorylated compounds that occur in living organisms [20,21,22]. Examples of such compounds are nucleotides, phosphoserine, phosphothreonine, phosphocholine, and phosphoethanolamine. Phosphoethanolamine (enP) plays a key role in the Kennedy pathway—the main metabolic route of synthesis phosphatidylethanolamine (PE) and phosphatidylcholine (PC)—the components of phospholipids which are structural and functional components of biomembranes [23,24,25]. Phosphoethanolamine and its derivatives have a significant influence on living organisms; they are useful for the treatment of cancer and infectious diseases, the tracking and prevention of some mutations, and antibiotic therapy directed to the bacterial membrane [26,27,28,29].

For phosphoethanolamine, the main coordination site in solution at a low pH value is the phosphate group, and at a higher pH value, the amine group becomes the metalation site [30]. Potential non-covalent interactions and metal-ion bonding sites for pyrimidine nucleotides are donor nitrogen atoms (N3) and phosphate groups [31,32,33].

This article presents the results of potentiometric and spectral studies of the complexes of phosphoethanolamine with copper(II) in ternary systems with monophosphorylated pyrimidine nucleotides: cytidine 5-monophosphate (CMP), uridine 5’monophosphate (UMP) and thymidine 5’-monophosphate (TMP).

## 2. Results and Discussion

The structures of the studied ligands are presented in Figure 1 and discussed with respect to the atom numbering shown in this picture. The values of the overall protonation constants of the ligands are given in Table 1.

For enP, CMP and UMP protonation constants and stability constants for their complexes with copper(II) ions in binary systems were previously described [32,34]. For TMP, we determined two protonation constants of thymidine 5-monophosphate by computer calculations from titration data. The first protonation constant for the N(3) atoms of TMP is log*K_1_* 9.73 (a similar value as for thymidine—log*K* 9.79 [32]), and it is assigned to the protonation of the endocyclic N(3) atom. The second log*K*_2_ is 6.04, which corresponds to the -O-PO_3_^2−^ group. The values of protonation constants for TMP as well as for UMP are much higher compared to CMP, which significantly changes the efficiency of metal ion binding. Deprotonation of the first proton of the phosphate group occurs at a relatively low pH value, and this protonation constant was not determined.

### 2.1. Cu/enP/TMP System

In the first step of the Cu(II)/enP/TMP study, the investigation in the binary system Cu(II)/TMP was performed. Potentiometric titrations in a metal:ligand ratio of 1:1 and the computer calculation of the potentiometric titration data was carried out taking into account the protonation constants of Table 1 and the constant for Cu(II) hydrolysis (log*β* = −13.13 for Cu(OH)_2_) [35]. The protonated form of the complex and hydroxocomplexes with their equilibrium constants were established based on the proposed reaction of their formation (ion charges were omitted for simplicity):Cu^2+^ + (HTMP) ⇆ Cu(HTMP) ⋯⋯⋯⋯⋯⋯log*K_e_* = 3.84
Cu^2+^ + TMP + H_2_O ⇆ Cu(TMP)(OH) + H^+^⋯⋯⋯⋯log*K_e_* = 14.16
Cu(TMP)(OH) + H_2_O ⇆ Cu(TMP)(OH)_2_ + H^+^⋯⋯log*K_e_* = 4.14

At the beginning of the measurement, free copper(II) ions were observed. From a pH close to 3.0, the first form of the complex Cu(HTMP) starts forming (Figure 2). It dominates at pH 6.0 and binds to around 55% of the total amount of copper ions in solution. From pH 5.5 to 11.0, the hydroxocomplex Cu(TMP)(OH) is observed, and it dominates at pH close to 8.0, binding over 85% of the copper ions. From pH 8.0, the concentration of the Cu(TMP)(OH)_2_ form begins to increase, and its dominant point is out of the measuring scale. At pH 11.0, it binds 85% of Cu^2+^. On the basis of the analysis of UV-Vis and EPR spectra, taking into account d-d transition energy as well as g_‖_ and A_‖_ parameters and changes in the chemical shifts of ^31^P and ^13^C NMR, the coordination mode was established. The Vis and EPR spectral parameters for the Cu(HTMP) complex *(λ*_max_ = 801 nm, g_‖_ = 2.37 and A_‖_ = 157 ∙ 10^−4^ cm^−1^, Table 2) indicate that only one oxygen atom is involved in coordination. Significant chemical shifts in the ^31^P and ^13^C NMR spectra of the ligand in complexes with respect to these on the free ligand on atom C(5’) indicate the activity of the phosphate group in the inner coordination sphere (^31^P −3.22 ppm, C(5’) 0.58 ppm) (Table 3). For the Cu(TMP)(OH) complex, the value of *λ*_max_ decreases to 711 nm, indicating the presence of nitrogen atom N(3) in the internal coordination sphere. Changes in the chemical shifts between free ligand and ligand in the complex in ^13^C NMR spectra (C(2) from 0.23 ppm to −0.08 ppm and C(4) from 0.05 to 0.12 ppm) prove this type of coordination. Changes in the chemical shifts on C(5’) atom (−0.72 ppm) show that the phosphate group is still an active site in the complexation process. A similar coordination mode with an additional oxygen atom from the hydroxyl group is observed for Cu(TMP)(OH)_2_ complex (*λ*_max_ = 673 nm). For this complex, EPR and NMR studies were impossible due to precipitation at samples made in higher concentrations.

In the copper(II)/phosphoethanolamine/thymidine 5’-monophosphate ternary system, complexes are formed (the overall stability constants are presented in Table 4 (Figure 2b). In this system, only one complex, Cu(enP)H_4_(TMP), dominates. At pH 2.5, this form binds almost 65% of total copper(II) ions introduced into the solution. According to the λ_max_ = 802 nm value, the EPR parameters of g_‖_ = 2.39 and A_‖_ = 135 ∙ 10^−4^ cm^−1^ (Table 5) and values of the protonation constants log*K_e_* = 5.70 and 6.04 for enP and TMP, respectively, and changes between chemical shifts on the ^31^P NMR spectra (−4.70 ppm for enP and −1.06 ppm for TMP) in the inner coordination sphere comprises only the oxygen atom of phosphate groups of enP (see Appendix A). That shift in the phosphorus atom of TMP may indicate an interaction with the amine group of enP. Analysis of the FT-IR spectrum confirmed these interactions (antisymmetric stretching band at 1079 cm^−1^ in the IR spectrum of the complex and at 1084 cm^−1^ in the spectrum of the free ligand [36]). From the beginning of the measurement to pH 7.0, the Cu(enP)H_3_(TMP) complex appears, the maximum concentration of which overlaps the range of domination of Cu(enP)H_4_(TMP), Cu(enP)H_2_(TMP) and Cu(HTMP), which makes a spectral study of this complex impossible to perform. A similar mode of interaction is observed for the Cu(enP)H_2_(TMP) complex (appears in the pH range 4.5 to 8.0) and Cu(enP(TMP)(OH)_2_. In this system, binary complexes are formed at a relatively high concentration.

### 2.2. Cu/enP/UMP System

The complexes forming in the ternary system Cu(II)/enP/UMP are Cu(enP)H_4_(UMP), Cu(enP)H_3_(UMP), Cu(enP)H_2_(UMP), Cu(enP)(UMP) and dinuclear mixed complex Cu_2_(enP)_2_(UMP). The first protonated complex Cu(enP)H_4_(UMP) binds almost 100% of copper(II) ions at the beginning of the measurement (Figure 3). The spectral results (Vis and EPR) indicate the formation of an {O_x_} chromophore (*λ_max_* = 798 nm, g_‖_ = 2.41 and A_‖_ = 137 ∙ 10^−4^ cm^−1^) (Table 5, Figure 4). As indicated by the changes in the NMR spectrum, in the inner coordination sphere, there is a phosphate group of phosphoethanolamine (^13^C NMR C(1) −0.94 ppm, ^31^P NMR −0.92 ppm) (Figure 5). The change between shifts on the second carbon of enP C(2) −0.94 ppm neighbouring the protonated amine may be a result of non-covalent interactions with the phosphate group of UMP as a negative center. For UV-Vis, EPR and NMR spectra, see Appendix A). This interaction was confirmed by the IR spectra of the complex related to the free ligand, where the positions of the IR stretching vibration bands (1083 cm^−1^ for free UMP and 1079 cm^−1^ for complex) are shifted slightly (Figure 5).

Cu(enP)H_3_(UMP) is observed between pH near 3.0 and 7.0 and dominates at pH 5.0, binding 60% of copper(II) ions introduced into the solution. Spectral parameters change—*λ_max_* decreases to 798 nm, g_‖_ = 2.40 and A_‖_ = 146 ∙ 10^−4^ cm^−1^. These small changes indicate the inclusion of another oxygen atom in the inner coordination sphere. The shift differences for ^31^P NMR for enP and UMP are −4.68 and −3.00 ppm, respectively, and they are much larger than for the Cu(enP)H_4_(UMP) complex. The shifts between the free ligand and the ligand in the complex of enP (C(1) 0.53 ppm, C(2) −0.07 ppm) and UMP (C(2) −0.02 ppm, C(4) 0.01 ppm, C(5’) −0.09) confirm the activity of the phosphates of the ligands and the absence of the activity of the endocyclic nitrogen atom N(3) of UMP.

At pH 6.4, Cu(enP)H_2_(UMP) complex binds 40% of copper ions. Between pH values of 5.0 and 8.0, it overlaps with Cu(enP)H_3_(UMP) and Cu_2_(enP)_2_(UMP) complexes, which makes spectral investigations impossible to perform.

From pH 6.0 to 11.0, the Cu_2_(enP)_2_(UMP) complex is observed and occurs at pH 7.1, binding around 70% of copper ions. This mixed-type dinuclear complex was previously reported for the system Cu(II)/CMP/OSpm [36]. The disappearance of the signal on EPR spectra confirms the formation of a dinuclear complex. Analysis of the energy of the d-d transitions indicates the {2N, xO}-type coordination (*λ_max_* = 698 nm). The shift differences in the signal positions coming from the carbon atoms neighbouring the nitrogen atom N(3) in the nucleotide are −0.04 and 0.08 ppm for C(2) and C(4), respectively, which are bigger than for Cu(enP)H_3_(UMP) and suggest the activity of this atom in coordination. Significant shifts for C = O on IR spectra were not observed. The shifts in phosphorus atoms decrease from −3.02 to −0.98 ppm and from −2.80 to −0.82 ppm for UMP and enP, respectively. These changes suggest copper ion is coordinated by the nitrogen atoms of the UMP and enP, and phosphate groups are still active in the metalation process.

At pH 9.5, Cu(enP)(UMP) is dominant, binding 80% of copper(II) ions. For this, complex spectral parameter from Vis spectroscopy (*λ_max_* = 685 nm) indicates the {2N, xO}-type coordination. As indicated by the changes in the ^13^C NMR and ^31^P NMR spectrum, copper ions are coordinated by the nitrogen atoms of the nucleotide (C(2) −0.42 and C(4) −0.24 ppm, P −0.03 ppm) and nitrogen atoms and oxygen atoms of the phosphate group of phosphothanolamine C(1) 0.69 and C(2) 0.46 ppm, P −1.10 ppm.

### 2.3. Cu/enP/CMP System

In the Cu(II)/enP/CMP system, Cu(enP)H_4_(CMP), Cu(enP)H_3_(CMP), Cu(enP)H_2_(CMP) Cu(enP)(CMP) and Cu(enP)(CMP)(OH) complexes were found (stability constants are given in Table 4). The first complex, Cu(enP)H_4_(CMP) (log*K_e_* = 10.79), existed in the system from the beginning of the measurements. It dominates at pH 2.5 and binds almost 100% of the copper(II) ions introduced into the solution (Figure 3). According to the *λ_max_* = 802 nm value, the EPR parameters of g_‖_ = 2.41 and A_‖_ = 141 ∙ 10^−4^ cm^−1^ (Table 5), values of the protonation constants log*K_e_* = 5.70 and 4.48 for enP and CMP, respectively, and chemical shifts between free ligand and ligand in the complex, on the ^31^P NMR spectra (−0.66 ppm for enP and −1.98 ppm for CMP), we concluded that in the inner coordination sphere, only the oxygen atoms of phosphate groups of CMP are involved (see Appendix A). Changes between chemical shifts of C(2) + 0.53 ppm on the ^13^C NMR spectra of CMP also indicate weak interactions between the carbonyl group of CMP as the negative center and the protonated amine group from phosphoethanolamine as the positive center.

With increasing pH value, Cu(enP)H_3_(CMP) starts to form and is dominant at pH = 5.2, binding more than 50% of copper ions. As follows from the d-d transition energy for this complex, the *λ_max_* = 792 nm value and the EPR parameters (g_‖_ = 2.35 and A_‖_ = 161 10^−4^ cm^−1^) (Table 5), the metalation involves two oxygen atoms. The changes in the chemical shifts between the free ligand and the ligand in the complex in the ^13^C NMR of enP (C(1) −0.05 ppm, C(2) 0.03 ppm) as well as in the ^31^P NMR (−2.91 ppm) confirm the coordination of the copper(II) ion with an oxygen donor atom of the phosphate group of phosphoethanolamine. The significant change in chemical shifts between free ligand and ligand in the complex in the ^31^P NMR spectrum of CMP (−3.98 ppm) indicates the participation of the phosphate group of CMP in coordination. Changes in the chemical shift on the ^13^C NMR (0.69 ppm) on the C(2) atom of CMP were observed due to the close proximity of the nitrogen N(3) atom. We excluded the participation of the carbonyl group in the complexation process due to the lack of shifts in the IR spectrum (1651 cm^−1^ for both the complex and free ligand).

The Cu(enP)H_2_(CMP) complex, created from pH 4.0 and dominant at pH 6.0, bound 65% of metal ions in the solution (Figure 3). The value of *λ_max_* = 751 nm and the EPR parameters (g_‖_ = 2.33 and A_‖_ = 159 ∙ 10^−4^ cm^−1^) (Table 5) indicate that in the inner coordination sphere, there are one nitrogen atom and oxygen atoms. The changes in the chemical shifts in the ^13^C NMR of enP (C(1) −0.97 ppm, C(2) −0.95 ppm) indicate that the phosphate group is still active in the coordination of copper ions. The changes in the chemical shifts between the free ligand and the ligand in the complex for CMP on the ^13^C NMR (C(2) 0.48 ppm) indicate that the nitrogen atom N(3) was in the inner coordination sphere.

The deprotonated Cu(enP)(CMP) complex was observed in the 5.7 to 11.0 pH range. It dominated at a pH close to 7.2 and bound almost all copper ions in the solution (Figure 3). For this complex, the formation of the stability constant (log*K_e_* = 14.50) is higher than for protonated forms and points to different modes of coordination. A significant decrease in the value of the maximum wavelength (*λ_max_* = 680 nm) and changes in the EPR parameters (g_‖_ = 2.30; A_‖_ = 158 ∙ 10^−4^ cm^−1^) means that in the inner coordination sphere, there are two nitrogen atoms. According to the ^13^C NMR and the ^31^P NMR spectra, we can observe decreasing changes in the chemical shift of the phosphate group of enP and C(1) atom as well as increasing changes in the chemical shift from the C(2) atom, which indicate the involvement of an amine group in complexation.

The hydroxocomplex Cu(enP)(CMP)(OH) begins to form at a pH close to 8.0 and becomes dominant at a pH of 10.0. An analysis of the Vis spectral studies (*λ_max_* = 689 nm) indicates that the inner coordination sphere is the same as in the Cu(enP)(CMP) complex, with the addition of one oxygen atom from the hydroxyl group. Because at pH 10.0, we can observe not only this ternary hydroxocomplex but also this binary Cu(enP)(OH)_2_ complex, this value of *λ_max_* may be overstated.

## 3. Materials and Methods

O-phosphoethanoloamine (enP), cytidine 5’monophosphate (CMP) and uridine 5’monophosphate disodium salt (UMP) were purchased from Sigma Aldrich (Steinheim am Albuch Baden-Württemberg, Germany), and thymidine 5’-monophosphate disodium salt (TMP) was purchased from Alfa Aesar (Thermo Fisher, Kandel, Germany) and used without additional purification. Copper(II) nitrate from Merck was purified by recrystallization from water. The concentration of copper ions in the solution was determined by the method of inductively coupled plasma optical emission spectrometry (ICP OES) (Shimadzu, Kyoto, Japan). All the prepared solutions and performed measurements were carried out with the use of demineralized, carbonate-free water.

Potentiometric titrations were performed using a Metrohm system (Titrino 702 equipped with an autoburette with a combined Metrohm glass electrode) (Metrohm AG, Herisau, Switzerland). Before each series of measurements, the pH meter indication was corrected with two standard buffer solutions of pH 4.002 and pH 9.225, and the electrode was calibrated in terms of H^+^ concentration [37]. The concentrations of phosphoethanolamine, nucleotides and copper(II) ions in the potentiometric studies were 1.0 × 10^−3,^ and the metal ion to ligand ratio was 1:1 for binary systems (Cu(II)/TMP) and 1:1:1 for ternary systems (Cu(II)/enP/NMP). All potentiometric titrations were carried out under strictly defined conditions: under a helium atmosphere (He 5.0) (Linde Gaz, Krakow, Poland), at a constant ionic strength of µ = 0.1 M (KNO_3_), a temperature 20 ± 1 °C, a pH range of 2.5 to 10.5, and with NaOH without CO_2_ at a concentration of 0.2011 M as a titrant. The calculations were performed using 150–350 experimental points for each titration. The data obtained from potentiometric titrations were subjected to computer analysis by the HYPERQUAD program for the determination of protonation constants and stability constants and the HYSS (Hyperquad Simulation and Speciation) program for the calculation of the distribution of particular species [38,39]. For the complexes formed in binary and ternary systems, the stability constants could be evaluated by the following equilibria (the charge was omitted for simplicity):$$\mathrm{pL}+{\mathrm{qH}}^{+}\;↔L_{p}H_{q}$$
$$\mathrm{pL}+qL^{\prime }+{\mathrm{rH}}^{+}\;↔L_{p}{L^{\prime }}_{q}H_{r}$$
$$\beta =\frac{\left[L_{p}L_{q}^{\prime }H_{r}\right]}{{\left[L\right]}^{p}{\left[\;L^{\prime }\right]}^{q}{\left[H\right]}^{r}}$$

The determined ionic product of water was pK_w_ = 13.78, and the hydrolysis constant for copper(II) was taken from our previous publications [40,41,42,43]. Testing began with the simplest hypothesis, and then the model was expanded to include progressively more complex forms [44]. After the improvement process, a set of complexes was established. The accuracy of the model was confirmed by verifying the results described in the papers [30,31,32].

^13^C and ^31^P NMR spectra were measured on an AVANCE III Bruker 500 MHz spectrometer (Bruker, Billerica, MA, USA). Dioxane as an internal standard for ^13^C NMR and phosphoric acid for ^31^P NMR were used. Samples for ^13^C and ^31^P NMR measurements were performed in D_2_O, and pD was adjusted by the addition of NaOD or DCl solutions, taking into account that pD = pH + 0.40 [45,46]. The concentration of the ligands in the samples was 0.1 M. The M:L molar ratio was 1:100 in binary systems, and the M:L:L’ molar ratio was 1:100:100 in ternary systems. The samples for infrared spectra were prepared in D_2_O; the value of M:L was 1:1, and the value of M:L:L’ was 1:1:1. Measurements were collected in the range of 400–4000 cm^−1^ using INVENIO R (Bruker, Bremen, Germany) with the ATR technique.

The UV-Vis spectra were determined using the Evolution 300 UV-VIS ThermoFisher Scientific spectrometer (Thermo Electron Scientific Instruments LLC, Madison, WI, USA) equipped with a xenon lamp (range 450–950 nm, accuracy 0.2 nm, and sweep rate 120 nm/min). Samples for the visible studies were prepared in H_2_O in molar ratios of 1:1 and 1:1:1 in concentrations C_M_ = 0.002–0.02 M. The spectra were recorded at 20 °C in a PLASTIBRAND PMMA cell (Brand, Wertheim, Germany) with a 1 cm path length.

The EPR spectra were recorded on a SE/X 2547 Radiopan instrument (Radiopan, Poznan, Poland). The samples were performed in a water:glycol mixture (3:1) at a concentration C_M_ = 0.005, and the measurements were carried out at −196 °C using glass capillary tubes (volume 130 µm^3^).

## 4. Conclusions

In all investigated systems, the formation of protonated complexes has been established, as well as MLL’ for Cu(II)/enP/CMP and Cu(II)/enP/UMP systems and the hydroxocomplexes MLL’(OH)_x_ for Cu(II)/enP/TMP and Cu(II)/enP/CMP. It should be noted that in the ternary system with uridine-5’-monophosphate, the mixed-type dinuclear complex Cu_2_(enP)_2_(UMP) is observed, which was confirmed by the disappearance of the signal in the EPR spectra. Analysis of the results presented above allows us to conclude that in the Cu(II)/enP/NMP systems, the reaction centers change depending on pH. At lower pH values, enP and nucleotides coordinate only via the phosphate groups, and with increasing basicity, the efficiency of phosphate groups decreases and the main reaction center becomes the amine group of enP and the endocyclic nitrogen atom N(3) from the pyrimidine ring of nucleotides. Additional non-covalent interactions have been found to occur between the bioligands where adducts are protonated amine groups from phosphoethanolamine and donor nitrogen atoms from the nucleotide. In binary systems, the stability of Cu(II)/NMP complexes increases in the order CMP < UMP < TMP, which corresponds to the values of their protonation constants. For ternary systems, with the presence of copper(II) ions, a nucleotide and a second ligand, which is phosphoethanolamine, we have also observed the lowest stability of ternary complexes of CMP, but for UMP and TMP, this order changes, and the same type complexes in the Cu(II)/enP/UMP system are more stable than complexes in the CU(II)/enP/TMP system. We hope that the results obtained will contribute to a broadening of the knowledge about complex compounds of phosphorylated ligands with copper (II) ions and their interaction with DNA and RNA. They may be crucial for explaining the processes taking place in living organisms, e.g., mutations and neurodegenerative diseases in the brain where phosphorylated lipids are abundant in the myelin sheaths.

## Figures and Tables

**Figure 1 ijms-23-13718-f001:**
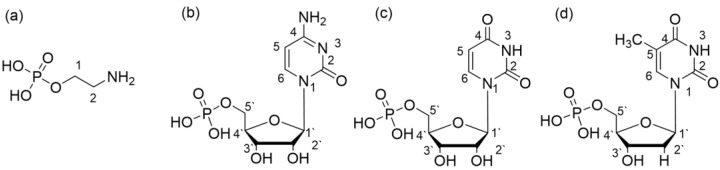
Chemical formula of the bioligand studied: (**a**) phosphoethanolamine (enP); (**b**) cytidine 5’-monophosphate (CMP); (**c**) uridine 5’-monophosphate (UMP); (**d**) thymidine 5’-monophosphate (TMP).

**Figure 2 ijms-23-13718-f002:**
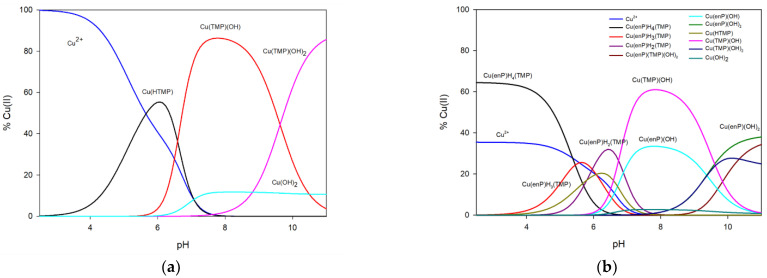
Distribution diagrams for (**a**) Cu(II)/TMP and (**b**) Cu/enP/TMP system (C_Cu_ = C_enP_ = C_TMP_ = 1.0 × 10^−3^ M).

**Figure 3 ijms-23-13718-f003:**
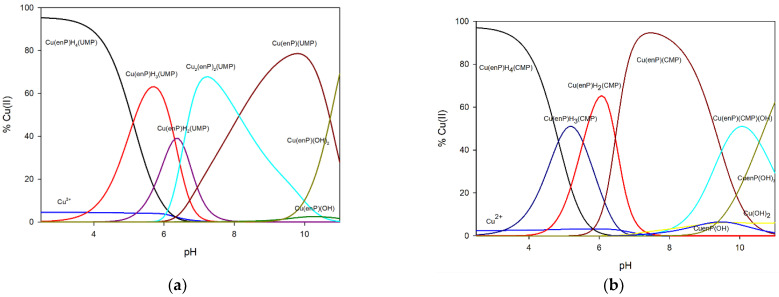
Distribution diagrams for (**a**) Cu(II)/enP/UMP and (**b**) Cu/enP/CMP systems.

**Figure 4 ijms-23-13718-f004:**
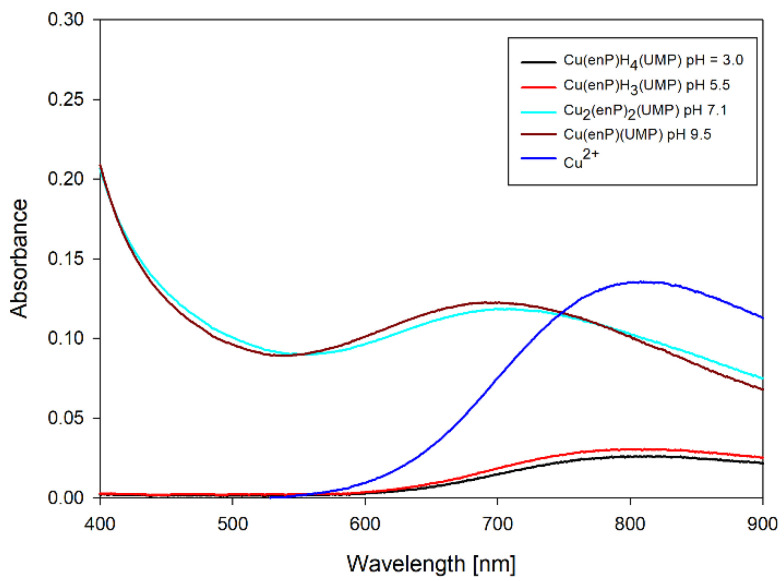
UV-Vis spectra of complexes in Cu(II)/enP/UMP system.

**Figure 5 ijms-23-13718-f005:**
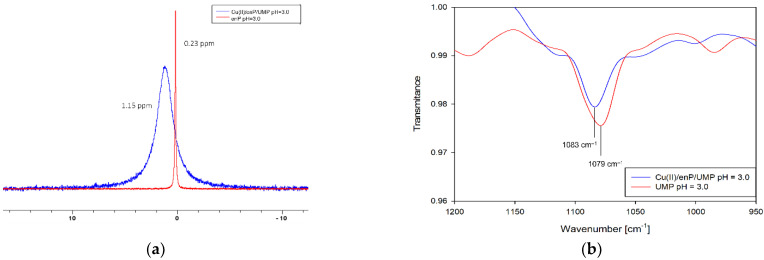
(**a**) ^31^P NMR spectrum of ligand and ligand in Cu(enP)H_4_(UMP) complex. (**b**) IR spectra of Cu(enP)H_4_(UMP) complex compared to the free ligand at the same pH.

**Table 1 ijms-23-13718-t001:** Protonation constants of ligands and stability constants for complexes in Cu(II)/L systems.

Species	enP [30]	TMP	UMP [34]	CMP [32]
H_2_L	16.11(3)	15.77(1)	15.13	10.90
HL	10.41(2)	9.73(1)	9.50	6.42
MHL	13.29(7)	13.57(3)	-	-
ML	-	-	6.03	2.71
ML(OH)	0.40(2)	0.39(2)	−2.82	−4.26
ML(OH)_2_	−7.26(4)	−9.24(2)	−13.02	-
ML(OH)_3_	-	-	−23.64	-

**Table 2 ijms-23-13718-t002:** Spectral parameters from UV-Vis and EPR studies for system Cu(II)/TMP.

Species	pH	*λ*_max_ (nm)	ε (dm^3^ mol^−1^ cm^−1^)	g_‖_	A_‖_ (10^−4^ cm^−1^)	Chromophore
Cu(HTMP)	6.0	801	19	2.37	157	{1 O}
Cu(TMP)(OH)	7.5	711	97		-	{1N, xO}
Cu(TMP)(OH)_2_	11.0	673	56	-	-	{1N, xO}

**Table 3 ijms-23-13718-t003:** NMR differences between signal positions for the ligand in the complex in relation to the free ligand (ppm).

		enP			Nucleotide										
System	pH	C1	C2	P	C2	C4	C5	C6	C1’	C2’	C3’	C4’	C5’	CH_3_	P
Cu(II)/TMP	6.0				+0.23	+0.05	−0.13	−0.16	+0.03	+0.22	−0.07	−0.28	+0.58	−0.15	−3.22
	7.5				−0.08	+0.12	+0.14	+0.16	−0.25	−0.26	+0.05	+0.23	−0.72	−0.10	−0.67
Cu(II)/enP/TMP	3.5	−0.85	−0.96	−4.70	0.00	+0.02	+0.01	−0.01	0.00	0.00	+0.02	0.00	−0.04	−0.01	−1.06
Cu(II)/enP/UMP	3.0	−0.94	−0.94	−0.92	+0.01	+0.01	0.00	0.00	0.00	0.00	0.00	0.00	−0.02		−0.98
	5.5	0.53	−0.07	−4.68	−0.02	+0.01	−0.02	+0.02	−0.04	0.01	0.02	+0.09	−0.09		−3.00
	6.4	−0.83	−1.00	−2.80	−0.04	0.08	−0.06	−0.01	−0.05	0.00	−0.01	−0.01	−0.04		−3.02
	7.1	−0.93	−0.98	−0.82	−0.15	+0.12	+0.07	0.00	−0.15	+0.30	−0.01	−0.01	+0.05		−0.98
	9.5	+0.69	+0.46	−1.10	−0.42	−0.24	+1.07	+0.05	−0.35	−0.03	−0.01	+0.01	−0.99		−0.03
Cu(II)/enP/CMP	2.5	−0.88	−0.88	−0.66	+0.53	+0.10	−1.65	−0.05	0.19	0.00	0.00	−0.02	−0.07		−1.98
	5.2	−0.05	+0.03	−2.91	+0.69	−0.14	-	+0.01	−0.63	−0.02	−0.02	−0.03	+0.04		−3.98
	6.0	−0.97	−0.94	−1.84	+0.48	−0.05	-	−0.01	−0.41	0.00	−0.02	−0.08	−0.01		−2.57
	7.2	−0.93	−0.95	−1.51	−0.06	−0.15	−0.44	−0.18	−0.04	−0.02	−0.15	−0.34	+0.42		−4.53
	10.0	−0.85	−1.34	−1.06	+0.99	+1.01	+0.98	+1.00	+0.97	+1.00	+1.00	+1.00	+0.98		−1.22

**Table 4 ijms-23-13718-t004:** Overall and stability constants as well as equilibrium constants of Cu(II) complexes in the Cu(II)/enP/NMP systems (standard deviation is given in parenthesis).

Species	Overall Stability Constants logβ	Reactions	log*K_e_*
Cu(enP)H_4_(TMP)	39.04(5)	Cu^2+^ + (H_2_enP) + (H_2_TMP) ⇆ Cu(enP)H_4_(TMP)	7.16
Cu(enP)H_3_(TMP)	33.59(5)	Cu(HenP) + (H_2_TMP) ⇆ Cu(enP)H_3_(TMP)	4.53
Cu(enP)H_2_(TMP)	27.63(4)	Cu(HenP) + (HTMP) ⇆ Cu(enP)H_2_(TMP)	4.61
Cu(enP)(TMP)(OH)_2_	−5.43	Cu(TMP)(OH) + enP + H_2_O ⇆ Cu(enP)(TMP)(OH)_2_	8.04
Cu(enP)H_4_(UMP)	41.27(2)	Cu^2+^ + (H_2_enP) + (H_2_UMP) ⇆ Cu(enP)H_4_(UMP)	10.03
Cu(enP)H_3_(UMP)	36.15(2)	Cu(HenP) + (H_2_UMP) ⇆ Cu(enP)H_3_(UMP)	7.73
Cu(enP)H_2_(UMP)	29.85(4)	Cu(HenP) + (HUMP) ⇆ Cu(enP)H_2_(UMP)	7.06
Cu_2_(enP)_2_(UMP)	29.94(4)	Cu(UMP) + 2 enP + Cu^2+^ ⇆ Cu_2_(enP)_2_(UMP)	23.91
Cu(enP)(UMP)	15.96(2)	Cu(UMP) + (enP) ⇆ Cu(enP)(UMP)	9.93
Cu(enP)H_4_(CMP)	37.80(2)	Cu^2+^ + (H_2_enP) + (H_2_CMP) ⇆ Cu(enP)H_4_(CMP)	10.79
Cu(enP)H_3_(CMP)	32.96(2)	Cu^2+^ + (HenP) + (H_2_CMP) ⇆ Cu(enP)H_3_(CMP)	11.65
Cu(enP)H_2_(CMP)	27.42(2)	Cu^2+^ + (HenP) + (HCMP) ⇆ Cu(enP)H_2_(CMP)	10.59
Cu(enP)(CMP)	14.50(2)	Cu^2+^ + (enP) + (CMP) ⇆ Cu(enP)(CMP)	14.50
Cu(enP)(CMP)(OH)	5.02(2)	Cu(CMP) + (enP) + H_2_O ⇆ Cu(enP)(CMP)(OH) + H^+^	16.17

**Table 5 ijms-23-13718-t005:** Spectral parameters from UV-Vis and EPR studies for ternary systems Cu(II)/enP/NMP.

Species	pH	λ_max_ (nm)	ε (dm^3^ mol^−1^ cm^−1^)	g_‖_	A_‖_ (10^−4^ cm^−1^)	Chromophore
Cu(enP)H_4_(TMP)	2.5	801	13.9	2.39	135	{1O}
Cu(enP)H_3_(TMP)	5.5	-	-	-	-	-
Cu(enP)H_2_(TMP)	6.5	-	-	-	-	-
Cu(enP)(TMP)(OH)_2_	11.0	-	-	-	-	-
Cu(enP)H_4_(UMP)	3.0	798	13.39	2.41	137	{1O}
Cu(enP)H_3_(UMP)	5.5	776	15.41	2.40	146	{2O}
Cu(enP)H_2_(UMP)	6.4	-	-	-	-	-
Cu_2_(enP)_2_(UMP)	7.1	698	62.89	-	-	{2N, xO}
Cu(enP)(UMP)	9.5	685	65.36	2.38	150	{2N, xO}
Cu(enP)H_4_(CMP)	2.5	802	11.75	2.41	141	{1O}
Cu(enP)H_3_(CMP)	5.2	792	14.43	2.35	161	{2O}
Cu(enP)H_2_(CMP)	6.0	751	40.79	2.33	159	{1N, xO}
Cu(enP)(CMP)	7.2	680	99.41	2.30	158	{2N, xO}
Cu(enP)(CMP)(OH)	10.0	689	77.73	-	-	{2N, xO}

## Data Availability

Not applicable.

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
