# Peer review of "Coordination Chemistry of Phosphate Groups in Systems Including Copper(II) Ions, Phosphoethanolamine and Pyrimidine Nucleotides"

_ijms, 2022, doi:10.3390/ijms232213718_

Round 1

Reviewer 1 Report

The manuscript by Gabryel-Skrodzka et al. describes the results of the potentiometric titration determination of the formation constants of copper(II)- thymidine 5`-monophosphate-coligand complexes. In addition, NMR, EPR and IR spectroscopy was used to support the findings of the research. The manuscript is well-written and only minor remarks may be given:

1. The results of NMR, IR and EPR spectroscopy are given in the tabular form. I suggest to also include the key experimental spectral plots into the supplementary material.

2. In the introduction the authors give an overview of the role of copper-phosphoethanolamine and other interactions in important biological processes. Can they also briefly explain in the Conclusions, how the results obtained may help in understanding of copper-mediated biochemical reactions?

3. The last part of the manuscript (technical) Institutional Review Board Statement and further contain text from the template and should be filled with the appropriate data or removed.

Author Response

The manuscript by Gabryel-Skrodzka et al. describes the results of the potentiometric titration determination of the formation constants of copper(II)- thymidine 5`-monophosphate-coligand complexes. In addition, NMR, EPR and IR spectroscopy was used to support the findings of the research. The manuscript is well-written and only minor remarks may be given:

  1. The results of NMR, IR and EPR spectroscopy are given in the tabular form. I suggest to also include the key experimental spectral plots into the supplementary material.

Ad 1. UV-Vis and EPR as well as NMR spectra have been added to the supplementary file. We chose the set of each spectra for one form of the complex in each MLL` system. The form which is existing in each system as dominant form is MLH4L`.

  1. In the introduction the authors give an overview of the role of copper-phosphoethanolamine and other interactions in important biological processes. Can they also briefly explain in the Conclusions, how the results obtained may help in understanding of copper-mediated biochemical reactions?

Ad 2. The short explanation has been added to the conclusions section.

  1. The last part of the manuscript (technical) Institutional Review Board Statement and further contain text from the template and should be filled with the appropriate data or removed.

Ad 3. Thank you for pointing out, some sections are updated and unnecessary ones are deleted.

Reviewer 2 Report

The article: “Coordination chemistry of phosphate groups in systems including copper(II) ions, phosphoethanolamine and pyrimidine nucleotides” presents interesting research on the process of copper complexes formation in the ternary systems with enP, CMP, UMP, and TMP.  As the authors mentioned in the paper, the interaction of divalent metal ions (from d-block) with DNA, nucleotides, and phosphate analogs are important topic nowadays. My overall pickup of this article is quite good. The methods and materials are properly described. The analytical methods used to study are very well selected and properly conducted, but some figures should better quality. The short introduction consists of the most important news and is well-grounded in the research topic, but I suggest you expand this paragraph. However, the manuscript needs a lot of corrections, which are listed below.

1.      The abstract is poor. It should be extended with more information about the most important achievements of the conducted research.

2.      I would expect a longer introduction, especially since there is a lot of work in the literature on the interactions of copper with nucleotides and phosphine compounds.

3.      The symbol of logarithmic value should be written in italic style font in all main text.

4.      The font legend and captions species distribution diagrams (Figure 2-3) are too small. It is really hard to read. So if it is possible please make them more readable. The same note to Figure 4 and 5.

5.      Please add the UV-Vis and EPR spectra plots in the supplementary file.

6.      Why you did not determine the value of AII for Cu(TMP)(OH)? This data is missing in Table 2. If gII was able to get the AII also, please add this dimension.

7.      The data in Table 3 are interesting, but in my opinion, it will be more useful to present them in a graphical way. However, I suggest including the NMR spectra in the manuscript or to the supplementary file.

8.      If you did not prepare the supplementary information file then you can delete line 316-317.

9.      In Conclusion paragraph, which is really short, it would be nice to add some structure of complexes. Moreover, expand the discussion and compare your result to similar complexes previously described in the literature, which is known to be very rich on this topic.

10.   Please check whole manuscript very carefully for linguistic and editorial errors, corrections are required in several places.

11.  The article was prepared hastily and carelessly on pages 9 and 10 there is a section with a description as in the template file. Of course, they should be replaced with an appropriate phrase and deleted when unnecessary, consequently: :’ Institutional Review Board Statement:’, , Informed Consent Statement:’, , Data Availability Statement:’, and ,Appendix A and B.

Due to the fact that the paper presents chemical research, I believe that the International Journal of Molecular Sciences is not the appropriate journal to publish this kind of study. I suggest editing the work according to comments and suggestions and considering submitting the article to Molecules (MDPI), which is much more suitable for the topic. There are currently open submissions to Special Issues with similar topics.

Author Response

The article: “Coordination chemistry of phosphate groups in systems including copper(II) ions, phosphoethanolamine and pyrimidine nucleotides” presents interesting research on the process of copper complexes formation in the ternary systems with enP, CMP, UMP, and TMP.  As the authors mentioned in the paper, the interaction of divalent metal ions (from d-block) with DNA, nucleotides, and phosphate analogs are important topic nowadays. My overall pickup of this article is quite good. The methods and materials are properly described. The analytical methods used to study are very well selected and properly conducted, but some figures should better quality. The short introduction consists of the most important news and is well-grounded in the research topic, but I suggest you expand this paragraph. However, the manuscript needs a lot of corrections, which are listed below.

  1. The abstract is poor. It should be extended with more information about the most important achievements of the conducted research.
  2. I would expect a longer introduction, especially since there is a lot of work in the literature on the interactions of copper with nucleotides and phosphine compounds.

Ad 1, ad 2. The abstract as well as introduction are extended with more information about the most important achievements of the conducted research.

  1. The symbol of logarithmic value should be written in italic style font in all main text.

Ad 3. Thank you for pointing out, symbols have been corrected throughout the text.

  1. The font legend and captions species distribution diagrams (Figure 2-3) are too small. It is really hard to read. So if it is possible please make them more readable. The same note to Figure 4 and 5.

Ad 4. We replaced figures with better quality files, we hope they are more readable now.

  1. Please add the UV-Vis and EPR spectra plots in the supplementary file.

Ad 5. UV-Vis and EPR as well as NMR spectra have been added to the supplementary file. We chose the set of each spectra for one form of the complex in each MLL` system. The form which is existing in each system as dominant form is MLH4L`

  1. Why you did not determine the value of AIIfor Cu(TMP)(OH)? This data is missing in Table 2. If gII was able to get the AII also, please add this dimension.

Ad 6. Thank you for your perceptiveness – we didn`t determine EPR spectral parameters for this form of the complex, the value of gII was the result of some mistake while copying data.

  1. The data in Table 3 are interesting, but in my opinion, it will be more useful to present them in a graphical way. However, I suggest including the NMR spectra in the manuscript or to the supplementary file.

Ad 7. Some of NMR spectra have been added to the supplementary file. We chose the set of each spectra for one form of the complex in each MLL` system. The form which is existing in each system as dominant form is MLH4L`.

  1. If you did not prepare the supplementary information file then you can delete line 316-317.

Ad 8. Supplementary material has been added and description has been updated.

  1. In Conclusion paragraph, which is really short, it would be nice to add some structure of complexes. Moreover, expand the discussion and compare your result to similar complexes previously described in the literature, which is known to be very rich on this topic.

Ad 9. The conclusions paragraph has been expanded in line with the comments.

  1. Please check whole manuscript very carefully for linguistic and editorial errors, corrections are required in several places.

Ad 10. The document has been improved in terms of linguistic proofreading.

  1. The article was prepared hastily and carelessly on pages 9 and 10 there is a section with a description as in the template file. Of course, they should be replaced with an appropriate phrase and deleted when unnecessary, consequently: :’ Institutional Review Board Statement:’, , Informed Consent Statement:’, , Data Availability Statement:’, and ,Appendix A and B.

Ad 11. Thank you for pointing out, some sections are updated and unnecessary ones are deleted.

Round 2

Reviewer 2 Report

Dear Authors,

Most of the points requiring the change have been improved. The quality of work in the current state is much better. Considering the above, I recommend the article "Coordination chemistry of phosphate groups in systems including copper(II) ions, phosphoethanolamine and pyrimidine nucleotides" written by Malwina Gabryel-Skrodzka and co-authors to publish in the International Journal of Molecular Science in the current version.